# Using deep mutational scanning to benchmark variant effect predictors and identify disease mutations

Benjamin J Livesey & Joseph A Marsh[*] (iD)

## Abstract

To deal with the huge number of novel protein-coding variants identified by genome and exome sequencing studies, many computational variant effect predictors (VEPs) have been developed. Such predictors are often trained and evaluated using different variant data sets, making a direct comparison between VEPs difficult. In this study, we use 31 previously published deep mutational scanning (DMS) experiments, which provide quantitative, independent phenotypic measurements for large numbers of single amino acid substitutions, in order to benchmark and compare 46 different VEPs. We also evaluate the ability of DMS measurements and VEPs to discriminate between pathogenic and benign missense variants. We find that DMS experiments tend to be superior to the top-ranking predictors, demonstrating the tremendous potential of DMS for identifying novel human disease mutations. Among the VEPs, DeepSequence clearly stood out, showing both the strongest correlations with DMS data and having the best ability to predict pathogenic mutations, which is especially remarkable given that it is an unsupervised method. We further recommend SNAP2, DEOGEN2, SNPs&GO, SuSPect and REVEL based upon their performance in these analyses.

**Keywords** missense mutations; phenotype prediction; protein structure; saturation mutagenesis; variant effect

**Subject Categories** Chromatin, Transcription & Genomics; Computational Biology

**Mol Syst Biol. (2020) 16: e9380**

## Introduction

Many genetic disorders can be attributed to sequence changes in protein-coding regions of DNA, yet pathogenic mutations account for only a tiny fraction of the overall genetic variation seen in humans. A typical pair of unrelated individuals will differ by approximately one nonsynonymous single nucleotide variant (SNV) per protein-coding gene (Rauch *et al*, 2012), while *de novo*

mutations lead to roughly one new nonsynonymous SNV per child not observed in either parent (de Ligt *et al*, 2012; Neale *et al*, 2012; Epi4K Consortium *et al*, 2013; Fitzgerald *et al*, 2015). The vast majority of mutations identified by sequencing are of unknown phenotypic consequence; that is, we are unsure if they have significant phenotypic effects or are functionally neutral. Thus, the ability to distinguish damaging variants from those that are benign is of tremendous importance for the diagnosis and treatment of human genetic disease.

In order to prioritise potentially pathogenic variants, many different computational variant effect predictors (VEPs) have been developed. These predictors make use of various protein sequence, structural, evolutionary and biophysical features to produce an effect score for the variant. By far the most commonly used feature is evolutionary sequence conservation and known variation (Table EV1). This is the only information used by several methods such as SIFT (Sim *et al*, 2012) and DeepSequence (Riesselman *et al*, 2018). Other predictors integrate additional features including biophysical properties of amino acids, protein functional annotations and epigenetic data (Rentzsch *et al*, 2019). Protein structural information, derived from experimentally determined models, is also used by several methods (Adzhubei *et al*, 2010; Capriotti & Altman, 2011), although there is conflicting information regarding whether its inclusion significantly improves predictor performance (Carraro *et al*, 2017).

While many of these approaches are able to make impressive predictions on test data sets and are widely applied in both clinical and research environments, there remain a number of unresolved sources of biases and inaccuracies. For example, when employing a supervised machine learning method, overfitting of the training set can become an issue. Instead of learning general rules, the predictor learns the niche peculiarities and noise of its training set (Srivastava *et al*, 2014). For this reason, machine learning techniques are usually subject to out-of-sample validation, whereby data not present in the training set are used to verify that the predictor has learned how to classify the data. Furthermore, when benchmarking these predictors with alternative data sets, they should contain as few mutations used during training and validation as possible. Biased representation within these data sets will skew the reported accuracy of methods trained and benchmarked with them (Schaafsma & Vihinen, 2018).

MRC Human Genetics Unit, Institute of Genetics and Molecular Medicine, University of Edinburgh, Edinburgh, UK
*Corresponding author. Tel: +44 (0) 1316 518538; E-mail: joseph.marsh@igmm.ed.ac.uk

Grimm *et al* (2015) describe two types of data circularity that can bias the assessment of predictor accuracy. Type 1 circularity occurs when the data from the training set are re-used for assessing predictor performance. This can occur due to overlap between commonly used variant databases. The result is a better apparent performance than if a more appropriate validation set were used. Metapredictors (trained using the outputs of other predictors) amplify this issue, as the methods they are built from often use different overlapping training sets. Type 2 circularity results in the weighting of predictor output by biases in the training examples. This can come about in VEPs due to ascertainment biases in the training set (long-studied proteins will have more annotated mutations than recently analysed ones). Another source is the association of certain genes with pathogenicity (e.g. many mutations in P53 will be damaging, while other genes may have no pathogenic mutations). Tools that use this information to weight their predictions can achieve excellent results on proteins with annotated pathogenic or benign mutations, but perform poorly when faced with unannotated proteins.

An alternative to computational predictions is experimental characterisation of mutation phenotypes. While this can be extremely time consuming if a separate experiment is required for each mutation, in recent years, an assortment of approaches has been developed for the high-throughput characterisation of mutation phenotypes. Deep mutational scanning (DMS) experiments combine saturation mutagenesis of a protein with a high-throughput functional test and deep sequencing (Fowler & Fields, 2014). The result is a framework, allowing the design of experiments to quantify the functional impact of a huge number of mutations at the same time. DMS experiments could potentially be hugely valuable for variant prioritisation, allowing direct identification of damaging human variants on a large scale (Majithia *et al*, 2016; Matreyek *et al*, 2018). DMS experiments can also be tailored to the specific definition of protein fitness required—something which computational methods are not able to account for (Harris *et al*, 2016). Even the best performing predictors struggle with more complex biological concepts such as allosteric regulation (Xu *et al*, 2017).

In addition to directly identifying damaging variants, another major benefit of DMS experiments is that they produce large variant effect data sets that can be used to benchmark and assess the performance of VEPs. These are fully independent from any training and testing data used by the predictors (with one exception Gray *et al*, 2018). Previous studies have found that using DMS data sets to benchmark computational predictors resulted in reduced predictive power compared to other commonly used data sets, suggesting that these predictors may not be as accurate for human variants as previously reported (Mahmood *et al*, 2017). The Critical Assessment of Genome Interpretation (CAGI) experiment, which aims to drive innovations in VEPs, frequently assesses predictors against novel unseen data sets (Hoskins *et al*, 2017) including those derived from DMS experiments.

In this study, we have taken advantage of the large number of DMS experiments that have now been published for a variety of diverse proteins from different organisms. First, we have used these data sets to perform an independent assessment and comparison of many different VEPs. Second, we have compared the ability of DMS experiments and VEPs to directly identify pathogenic human mutations.

# Results

## Overview of DMS data sets and variant effect predictors used in this study

To identify DMS data sets, we performed a literature search for papers presenting such experiments with available data. Using search terms such as "deep mutational scan", "fitness landscape", "massively parallel mutagenesis" and "saturation mutagenesis", we identified 31 viable DMS data sets (Table EV2). As shown in Fig 1, human proteins were the most numerous targets for these DMS experiments. *Saccharomyces cerevisiae* and *Escherichia coli* were also highly represented as they endogenously produce a number of model proteins, are easy to culture and maintain and are amenable to several effective assays for protein activity (e.g. growth rate and two hybrid). Proteins from viruses were also represented from studies investigating viral adaptation through massively parallel mutagenesis techniques.

There was considerable variation in functional assays applied between the DMS studies. Growth rate of yeast was the most common technique and was applied to several human proteins by knocking out the yeast orthologue and replacing it with the human gene that is capable of rescuing the null strain (Weile *et al*, 2017). Viral replication assays, performed by quantitative sequencing after a certain time point, were applied to all of the viral proteins (Wu *et al*, 2015; Doud & Bloom, 2016; Haddox *et al*, 2016; Lee *et al*, 2018). Survival assays involved placing the organism in hostile conditions where the target protein confers an advantage such as antibiotic resistance (Deng *et al*, 2012; Jacquier *et al*, 2013; Firnberg *et al*, 2014; Stiffler *et al*, 2015; Dandage *et al*, 2018). Two-hybrid assays allow protein–protein interactions to be analysed, while fluorescence can be used to investigate enzyme activity, protein stability or transcriptional pathway activation (Kitzman *et al*, 2015; Starita *et al*, 2015; Bandaru *et al*, 2017). Phage-display assays allow a number of protein attributes to be tested *ex vivo* by externalising the protein of interest followed by selection based on its attributes (Starita *et al*, 2015). The *E. coli* toxin ccdB was assayed by reverse survival, investigating its ability to restrict cell growth (Adkar *et al*, 2012).

Each study also varied in the coverage of possible single amino acid substitutions across the entire protein (Fig 1). Many of the studies included only those mutations that were possible by introducing a single nucleotide change, reducing potential coverage of all possible amino acid substitutions by around 70%. Some studies focused on specific regions of the target protein. In addition, most studies excluded low confidence mutants from their data, that is those with exceptionally low sequencing counts. For inclusion in this analysis, we required at least 5% coverage of all possible mutations in order to prevent unrepresentative low coverage data from skewing the results.

The computational VEPs used in this study were found using a number of approaches, including the OMICtools database (Henry *et al*, 2014), identifying tools tagged with "variant effect prediction" and searching for "protein variant effect prediction" and "protein phenotype predictor" using standard internet search engines. Priority was given to tools that featured either a web interface or an API that could be queried for thousands of mutations simultaneously. We also made use of the dbNSFP (Liu *et al*, 2016) database of pre-calculated predictions from multiple VEPs for the human genome (downloaded 2020-02-12). We split the predictors into four broad categories, based on the features that they use to make predictions:

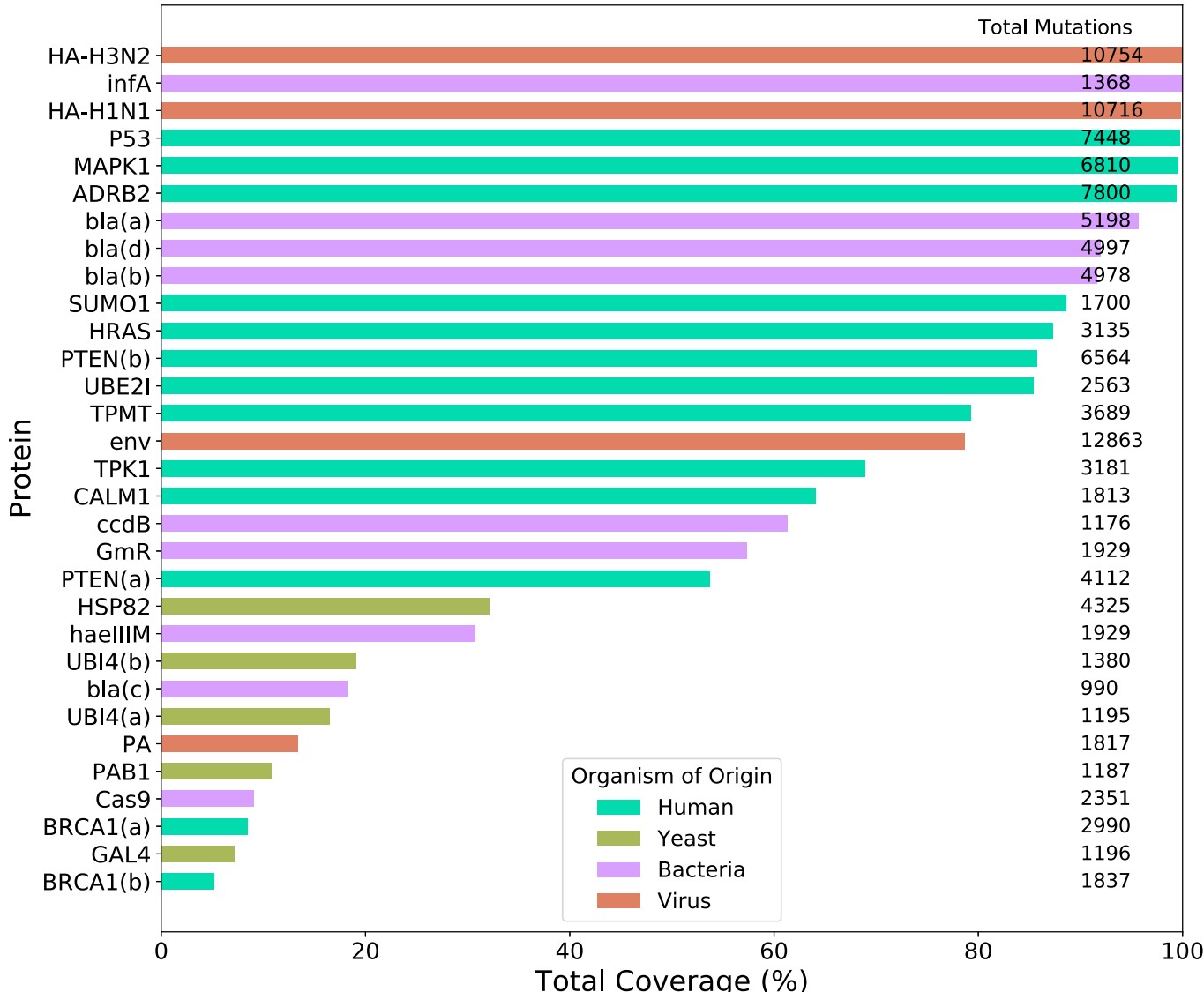

**Figure 1. Coverage of possible mutations by deep mutational scanning (DMS) experiments.**

The percentage of all amino acid substitutions covered by each DMS experiment. The total number of mutations assessed by each DMS experiment is indicated on the right. Where multiple data sets exist for a single protein, sequential letters are used to identify them.

1   *Supervised predictors*. These predictors use a machine learning technique that relies on learning from labelled examples, in particular data sets of known or suspected pathogenic and benign variants. Different predictors make use of a variety of different machine learning approaches, for example support vector machines and random forest algorithms.

2   *Unsupervised predictors*. These predictors make use of an unsupervised machine learning technique; that is, they are not trained using labelled pathogenic and benign variants. Instead, they rely mostly on evolutionary conservation from multiple sequence alignments. This includes unsupervised clustering techniques, hidden Markov models and generative models.

3   *Empirical predictors*. These predictors do not make use of any machine learning techniques, instead making an empirical calculation using the input data. This category also includes

amino acid substitution matrices and many evolutionary conservation metrics. Along with the unsupervised predictors, they should be free from any training bias.

4   *Metapredictors*. These predictors integrate other VEP results as input features, although many also use additional features. The metapredictors used in our study are nearly all trained using a supervised learning approach, with one exception (Ionita-Laza *et al*, 2016). To qualify for this category in our classification, a predictor must include at least two other VEPs as input features, not including substitution matrices or simple conservation metrics (such as GERP, PhyloP or SiPhy).

Among the DMS data sets, there are several instances of the same protein being investigated in different studies by different groups. Specifically, there are four independent data sets for β-lactamase (bla) and two each for UBI4, PTEN and BRCA1. There are also

two data sets for the influenza protein HA, but these were from different strains, so not directly comparable. To assess the reproducibility of DMS and its viability as a benchmark, we calculated Spearman's correlation coefficient between the functional scores of each DMS set in the same protein. Our results (Table EV3) demonstrate a range of correlations from 0.94 (bla(a)/(b)) to 0.34 (PTEN (a)/(b)). The average correlations observed over all pairs of analyses were 0.66. Some level of variance is expected due to differences in experimental method, fitness assays and conditions between experiments. Overall, the moderately high correlations suggest that DMS scores constitute a reasonably robust benchmark despite differing experimental conditions. We can also treat this correlation as a rough guide for how well we could expect a "perfect" computational predictor to perform against DMS data from these experiments.

We also assessed the correlation between different DMS data sets generated by the same studies. The purposes of these assays varied and included controls with no selection pressure, biological replicates, incrementally differing conditions and different fitness assays (Table EV4). Incremental changes in conditions tended to result in high correlations while larger alterations to conditions, assay type or protein partners resulted in much lower correlations between the data sets. Nonselective controls produced low correlations, while comparison of positive to negative selection assays produced a negative correlation. These results indicate that interpretation of DMS results depends to some extent on the exact fitness assay. However, certain mutations (e.g. those that destabilise the protein) are likely to always have an impact on fitness if a selection pressure is present.

## Assessment of variant effect predictors using DMS data

Where possible, we applied every computational predictor to each protein in the DMS data sets, substituting every possible amino acid at all positions. Some predictors failed to generate results for some proteins; this can occur due to an insufficiently deep multiple sequence alignment, mapping errors or other causes depending on the predictor. In order to get a measure of relative performance for each predictor, we calculated Spearman's rank correlation between the independent DMS scores for each protein and the predictions of every VEP (Fig 2). We also performed the same analysis using Kendall's Tau (Fig EV1) which produced only minor changes in predictor ranking and lower average correlations.

Given the large number of predictors that are specific to humans, we split this analysis up into human (Fig 2A) and non-human (Fig 2B) proteins. The top-performing predictor for each protein is labelled on the plot, while the full set of correlations are provided in Tables EV5 and EV6. Table EV7 shows the relative ranking of each predictor using a rank score that combines the rankings for all proteins in the human, yeast, bacterial and viral data sets (Table EV8 shows the same using Kendall's Tau instead).

DeepSequence was the overall top-performing method for predicting DMS results in the human proteins, showing the highest correlations out of all predictors for ADRB2, CALM1 and PTEN(b), and ranking within the top five predictors for seven of the 13 DMS data sets. It also had by far the highest rank score. To assess the statistical significance of this, we used a bootstrapping approach

and re-calculated the ranking by re-sampling all DMS data sets with replacement 1000 times. Strikingly, we found that DeepSequence always ranked the highest, showing that it is statistically better than all other predictors ($P < 0.001$). DeepSequence also ranked best for bacterial proteins, being the top predictor for six proteins: infA, bla (b), bla(c), ccdB, haeIIIM and GmR. In contrast, DeepSequence produced only a moderate rank score for yeast proteins, with a high coefficient of variation. This was largely due to poor performance on the ubiquitin (UBI4) data sets, which reduced the overall rank score considerably. If the UBI4 data sets are excluded from the analysis, then DeepSequence becomes the second-highest ranked predictor for yeast proteins. Interestingly, DeepSequence performs poorly for viral proteins, ranking second to last out of all predictors tested. This is consistent with the original publication, where the creators report poor performance on viral proteins due to insufficient sequence diversity within the alignments used (Riesselman *et al*, 2018).

The authors of DeepSequence tested their method using DMS data sets, including many that are included in our study, so it is conceivable that method hyperparameters may have been optimised to provide an advantage predicting data from these specific proteins. To address this potential, we repeated the ranking process using only DMS data sets from proteins that were not included in the original DeepSequence study (Table EV9). Importantly, DeepSequence still ranks top among all predictors, suggesting that its apparent success is not simply due to hyperparameter optimisation.

Among the other predictors, certain supervised approaches were particularly notable. SNPs&GO (Capriotti *et al*, 2013) ranked 2[nd] for human and 1[st] for yeast proteins, although its predictions were relatively poor for non-eukaryotic (bacterial and viral) proteins. SNAP2 (Hecht *et al*, 2015) also performed well, ranking 3[rd] for human proteins and 2[nd] for yeast. DEOGEN2 (Raimondi *et al*, 2017), a human-specific predictor, came 4[th]. SuSPect (Yates *et al*, 2014) showed good performance across most groups, ranking 5[th] for humans 3[rd] for bacteria and 2[nd] for viral proteins. REVEL was the only metapredictor to show notable performance, ranking 6[th] overall and having the highest correlation with the SUMO1 DMS data.

Some predictors incorporate features derived from experimentally determined protein structures into their predictions. Specifically, SNP&GOs3D and S3D-PROF (Capriotti & Altman, 2011) require a PDB structure to be provided in order to make their predictions and use features representing the 3D environment of the mutation. Other predictors such as PolyPhen-2 (Adzhubei *et al*, 2010), DEOGEN2 and MPC derive some features from experimental structures, but are still capable of making predictions without them. While these methods ranked average-to-high and achieved a number of top correlations with the DMS data, overall they do not perform better than the top-performing sequence-based methods. We also find that, in proteins with partial coverage of high-resolution structures, the difference in performance between areas of structural coverage and areas with no coverage is comparable between structural methods and pure sequence-based methods (Fig EV2). This may be due to regions without structures being more likely to be disordered and less conserved, and thus harder to characterise by conservation metrics.

Of all predictors, FATHMM (Shihab *et al*, 2013) produced the most significant outlier, generating predictions with by far the highest correlation for P53, but having low correlations for all other

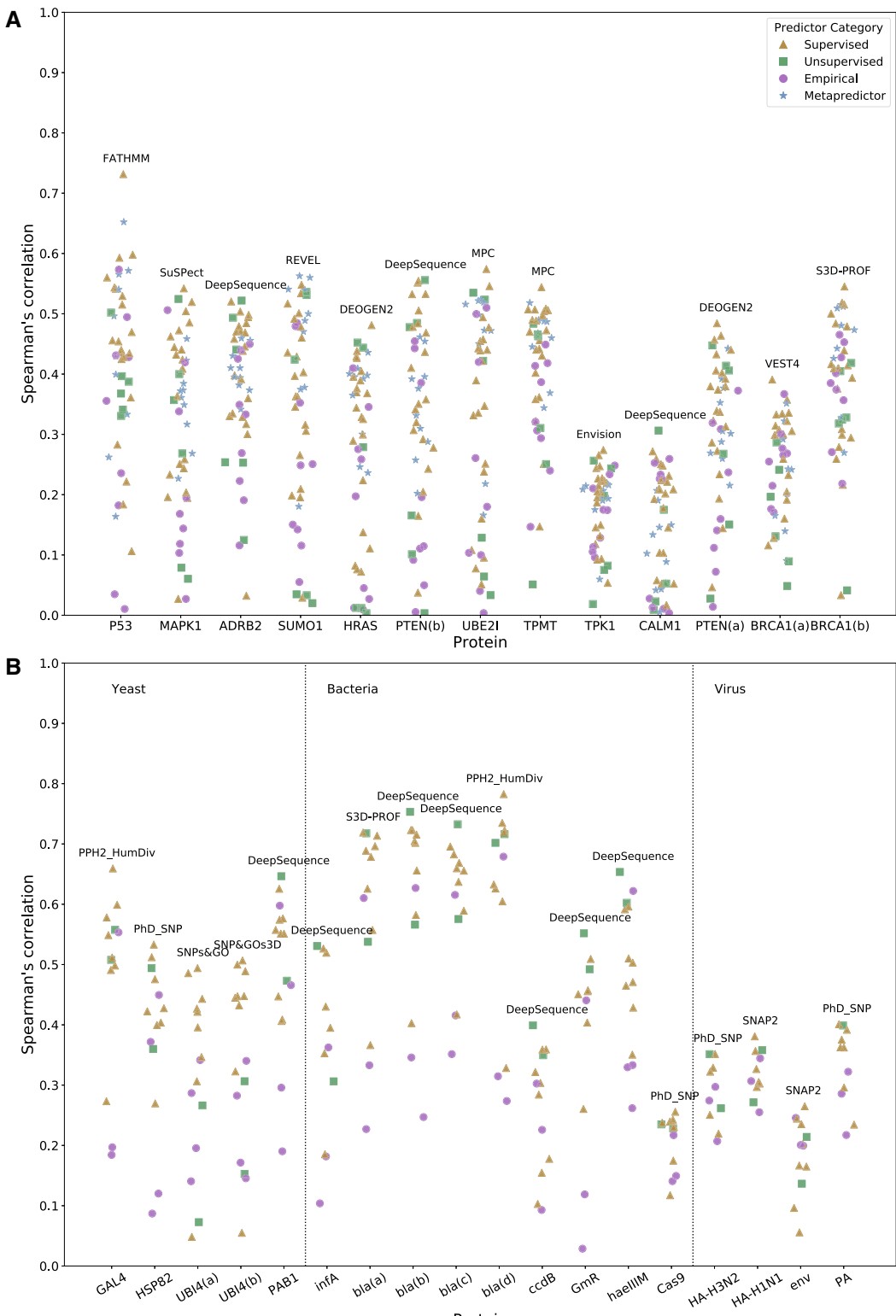

**Figure 2. Correlations between computational variant effect predictors and deep mutational scanning (DMS) measurements.**

A, B    Spearman's correlation calculated between all variant effect predictions and DMS data sets. The top-performing predictor for each protein is labelled on the plot. This analysis is split into (A) human and (B) non-human proteins.

proteins, resulting in an overall low-rank score with a high coefficient of variation. The explanation for this is unclear, but it may be due to overfitting of the predictor for specific proteins, given the enrichment of P53 mutations in human disease databases compared to many of the other proteins in this study.

Different DMS data sets varied greatly in their correlations with the computational predictors. In particular, BRCA1, CALM1 and TPK1 among the human proteins and ccdB, Cas9 and env among the non-human proteins showed low correlations, even from the best predictors. As far as we can tell, this effect appears to be unrelated to protein coverage, data set size or experimental methodology. For example, UBE2I, SUMO1, TPK1 and CALM1 were all studied by the same group using the same approach (growth rate in yeast) (Weile *et al*, 2017), yet UBE2I and SUMO1 show markedly higher correlations with all predictors than the others. Viral proteins also showed low correlations, and in fact, the simple BLOSUM62 substitution matrix (Henikoff & Henikoff, 1992) was the most highly correlated with the env data set when using Kendall's Tau (second highest when using Spearman's). This indicates that the inclusion of typical training features is of less use when predicting the fitness of viral proteins, likely due to lack of viral representation in training sets and lack of viral sequence diversity in many databases used to generate multiple sequence alignments. Viral proteins may also be more likely to undergo adaptive evolution, thus potentially confounding conservation-based approaches.

It is also interesting to note that, despite the fact that most of the predictors used in this study are human-specific, the top-ranking predictors for the human DMS data sets tend to be general predictors applicable to proteins from all species. For example, for the human DMS data sets, only one of the top five predictors is specific to humans whereas many of the lowest ranked predictors are human-specific. An important contributing factor to this may be overfitting against human mutation data sets for some predictors, which causes them to perform poorly against independent experimental phenotype measurements. In addition, several of the worst predictors are also based upon nucleotide-level constraint (GERP++ (Davydov *et al*, 2010), SiPhy (Garber *et al*, 2009), phastCons (Siepel & Haussler, 2005) and fitCons (Gulko *et al*, 2015)). These predictors ranked even lower than the simple BLOSUM62 and Grantham (Grantham, 1974) substitution matrices, suggesting that such approaches are poorly suited to predicting the protein-level effects of mutations.

Many DMS experiments included amino acid substitutions that are not possible by single nucleotide changes; that is, they are technically not missense variants. Some VEPs do not produce predictions for these mutations, particularly those that take nucleotide-level substitutions into consideration. Therefore, we repeated our analysis, limiting our predictions to only those amino acid substitutions that are possible by single nucleotide changes (Table EV10). The rankings remain broadly similar, and the top-ranking method did not change for any group.

## Identification of pathogenic human mutations using DMS data and computational variant effect predictors

We next investigated the ability of both DMS experiments and VEPs to distinguish pathogenic human missense mutations, taken from the ClinVar database (Landrum *et al*, 2014), from missense variants observed in the human population, taken from gnomAD (preprint: Karczewski *et al*, 2019). While some gnomAD variants may be damaging under certain circumstances (e.g. if associated with recessive, late-onset or incomplete penetrance disease), we assume that the vast majority of them should be non-pathogenic, and therefore refer to them as "putatively benign". Of the 11 human proteins with DMS data sets, seven have known pathogenic or likely pathogenic missense variants in ClinVar as of 2019-10-25 (93 for BRCA1, 31 for HRAS, 189 for P53, 108 for PTEN, nine for CALM1, five for TPK1 and two for MAPK1). For CALM1 and TPK1, we identified additional pathogenic missense mutations in the literature (Crotti *et al*, 2013; Banka *et al*, 2014; Jensen *et al*, 2018; Nomikos *et al*, 2018; Zhu *et al*, 2019), leading to a total of 19 for CALM1 and eight for TPK1. MAPK1 has too few recorded pathogenic missense variants to include in this analysis.

For each predictor, we plotted a receiver operating characteristic (ROC) curve for classification performance on every protein, identifying pathogenic ClinVar mutations as true positives and the putatively benign gnomAD mutations as true negatives (removing any ClinVar mutations from the gnomAD set). We then calculated the area under the curve (AUC) for each plot as a measure of that predictor's performance in classifying the data (Fig 3). We also calculated the precision–recall AUCs (Fig EV3). Descriptions of each DMS data set displayed in Fig 3 are provided in Table EV11.

In the ROC analysis, an experimental DMS metric performed better than any of the 46 VEPs for four of the six human proteins (Fig 3A,B,D and E) and ranked relatively high for the remaining two (Fig 3C and F: CALM1 and PTEN). To determine the significance of the performance of the DMS data, we used a bootstrapping approach and individually re-sampled the gnomAD and ClinVar data sets with replacement 10,000 times, re-calculating the AUC scores with the new data. DMS ranked first in 9202 trials, while DeepSequence came top in 600 trials, REVEL in 154, MutPred in 34, SNPs&GO in 7, SIFT4G in 2 and PhD-SNP in 1. Thus, while we cannot quite state at this point that DMS is significantly better than all computational predictors together ($P = 0.080$), it clearly ranks higher than all VEPs in our analysis and is significantly better than all except DeepSequence. Very similar results are observed for the precision–recall AUCs (Fig EV3), except that the TPK1 DMS data sets changed from ranking $1^{st}$ to $6^{th}$.

The DMS results for CALM1 and TPK1 were generated by the same group using the same method, assessing the effects of mutations on growth rate in a yeast system (Weile *et al*, 2017). The data processing pipeline used in this study penalised "hypercomplementing" variants (i.e. those with fitness greater than the wild type) by setting the fitness to the reciprocal of the measured value. These are labelled as "flipped" in Fig 3. Interestingly, we found that while these "flipped" DMS results show a better correlation with the outputs of VEPs than the raw DMS data (Tables EV12 and EV13), the raw scores are better for directly identifying pathogenic variants (Fig 3). This suggests that VEPs in general tend to be predictive of a perturbation away from wild-type activity (regardless of whether it is an increase or decrease), whereas only a decrease in activity is predictive of disease, at least for these two proteins. This is consistent with a recent observation that beneficial effects on protein function, as measured by DMS experiments, were predicted less well than detrimental effects for all four tested VEPs (preprint: Reeb *et al*, 2019).

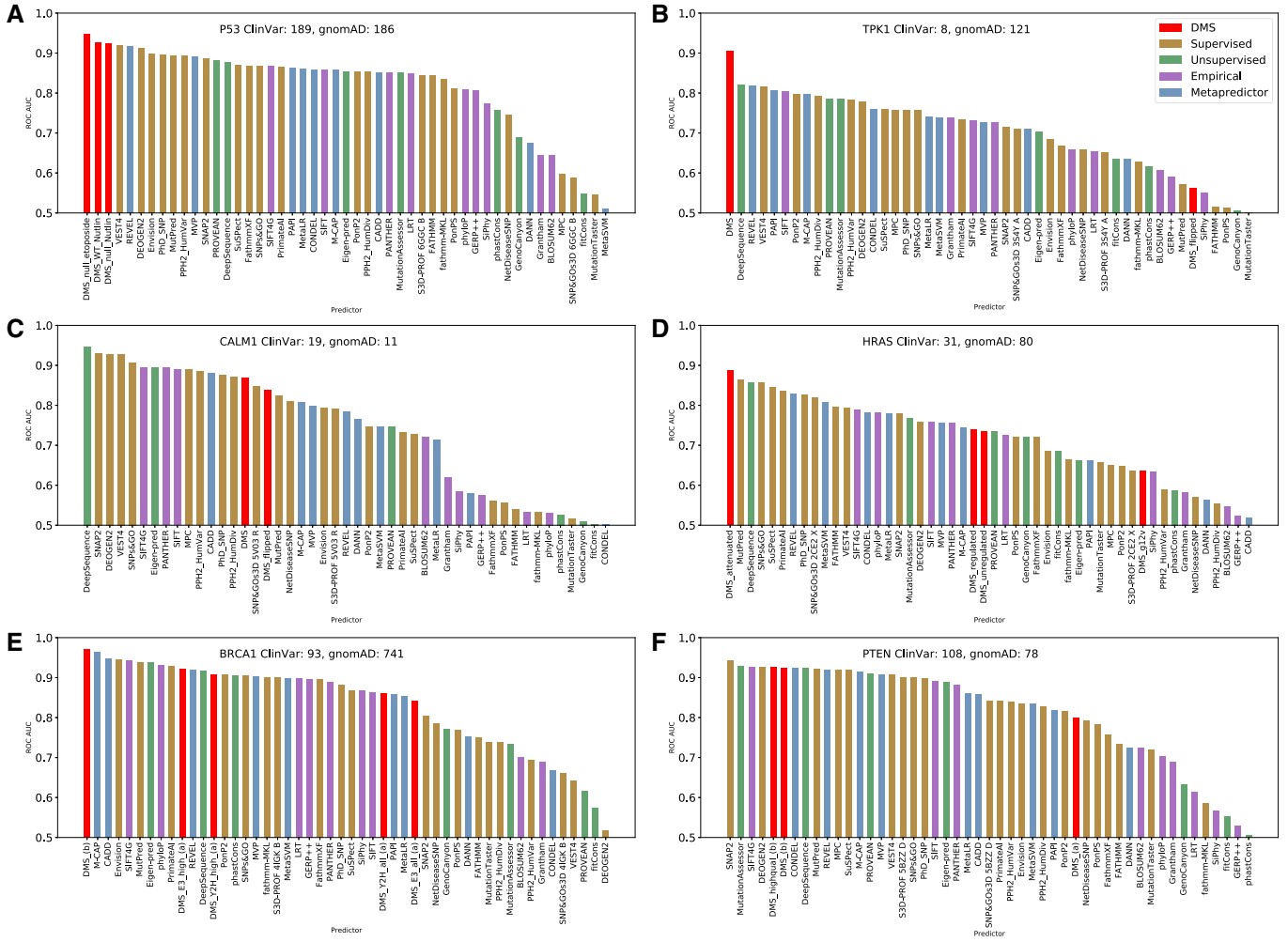

**Figure 3. Identification of pathogenic missense variants by deep mutational scanning (DMS) data sets and variant effect predictors (VEPs).**

A-F   ROC AUC values for DMS data sets and VEPs in distinguishing between pathogenic missense variants from ClinVar and putatively benign variants from gnomAD for six human disease genes (A–F). The numbers of variants in each class are indicated on the plot. The different DMS data sets for each protein are described in Table EV7.

While the primary objective of this analysis is to compare the DMS data sets to the VEPs, it is also interesting to observe the relative performances of the different computational predictors in terms of directly identifying pathogenic mutations for the six human proteins in Fig 3. This comparison is limited by the fact that there is likely some overlap between the mutations used to evaluate the predictors here, and the mutations originally used to train some of the supervised predictors and metapredictors. In this regard, it is especially interesting to see that the unsupervised predictor DeepSequence again stood out among VEPs, ranking 1st for TPK1 and CALM1, 2nd for HRAS, and within the top 11 predictors for all remaining proteins. This is considerably better and more consistent performance than any of the other computational predictors. A few other VEPs also performed well, but these are dominated by supervised predictors and metapredictors. Given that the training of these methods almost certainly included many of the mutations used in this analysis, ranking the relative performance of these methods will be heavily subject to any training bias and beyond the scope of this study to assess.

We did notice that certain predictors performed particularly poorly on certain targets. For example, DEOGEN2 ranked last, by far, for BRCA1. Interestingly, however, the relative performance of DEOGEN2 improved markedly if only predictions of mutations with DMS measurements, which covers primarily just the RING domain of BRCA1, are considered (Fig EV4). This appears to be due to DEOGEN2 assigning extremely different weights to different domains in BRCA1, thus obscuring good predictions when analysing the entire protein. We also investigated other predictors with low AUCs for additional domain-specific effects. A further three data sets which showed a similar pattern were MPC on P53, VEST4 on BRCA1 and PROVEAN on BRCA1, which are all highlighted in Fig EV4.

## Discussion

The number of available genome and protein sequences has increased tremendously in the last decade due to advances in next-

generation sequencing technologies. In this wealth of new data, we have discovered a large number of previously unseen coding variants of unknown functional significance. To assist us in analysing this new data, computational predictors have been developed, but the training and evaluation of these predictors often suffer from biases. DMS experiments provide an ideal benchmark for testing predictors, ensuring that none of the training data is included in the evaluation. The availability of a large number of such experimental DMS data sets has facilitated this study.

We are aware that numerous technical and computational factors can impact the quality of data from DMS studies. These can stem from experimental procedure and thus be assessed through reproducibility in biological replicates, or measurement uncertainty assessed by technical replicates. The largest source of error from DMS is encountered in the sequencing stage, where next-generation sequencing typically reads between 1/100 and 1/1,000 bases incorrectly (Ma *et al*, 2019). Many groups adopt a barcoding strategy to address this issue, so that a multi-base unique artificial sequence is associated with each variant. In addition, reads below a certain quality threshold are rejected and variants which are present at a rate below a given detection threshold are removed. Several groups provide both their full fitness scores and a filter for high-quality results (Starita *et al*, 2015; Mighell *et al*, 2018). In these cases, we find that the filtered high-quality results have a higher average correlation with the VEPs (Tables EV12 and EV13), as well as superior predictive power for disease mutations (Fig 3).

Of the 46 different predictors evaluated in this study, we find that a single program, DeepSequence, clearly stands out from all of the others, both in terms of performance and in terms of methodology. DeepSequence showed the strongest correlations with the DMS data in humans and bacteria and was the top computational predictor of human disease mutations. Most machine learning methods make use of several features, often including some measure of sequence conservation at the site of interest, and then learn the patterns of these features that result in a mutation being classified as damaging or benign. DeepSequence makes use of deep generative models to integrate factors from the entire sequence at once, rather than only one or a few sites. This type of problem is largely intractable for traditional machine learning, given the number of parameters involved; however, DeepSequence overcomes this by learning the latent factors underlying the protein sequence. This approach also produces advantages in terms of the biases inherent in supervised methods. We can expect a machine learning method confronted with an example it was trained on to classify it correctly most of the time, producing an unrepresentative assessment of its accuracy. DeepSequence makes use of multiple sequence alignments and never sees labelled protein data, resulting in scores that are not biased by training examples. This is not to say that DeepSequence is a completely unbiased method, however. The scores which are generated depend entirely upon the database from which multiple sequence alignments are drawn. If certain sequences are under-represented, then predictions for those will be lower quality, such as the results we observe for viral proteins drawn from the UniRef100 database. The success that DeepSequence has achieved in predicting mutation effects for human proteins shows that deep generative models may well be the way forward in this field, removing the reliance on labelled data sets for making predictions.

One of the VEPs we assessed, Envision, is trained with a supervised learning approach using DMS data rather than labelled pathogenic and benign variants. This method uses a number of the same DMS sets we used in this analysis for training (BRCA1 (a), HSP82, UBI4(a and b), PAB1 and bla(a)); thus, the ranking of this method in Table EV7 is almost certainly subject to training bias. It is interesting, however, that despite this advantage, Envision only produces moderate overall performance for human DMS data sets (although it does rank 1st for TPK1). In terms of predicting pathogenic missense mutations, Envision performs well for BRCA1 ranking 3rd among the VEPs and P53, ranking 4th, but its performance is unremarkable for the other proteins. Notably, although Envision was not trained on a P53 data set, it was evaluated using one (although not the same DMS data set used in this study). While the approach used by Envision is innovative, assessing its performance with DMS has the same caveats as assessing performance of other supervised VEPs using pathogenic mutation databases. Thus, it is notable that, despite this advantage, Envision showed only modest performance against the DMS data.

Most predictors, supervised or otherwise, undergo hyperparameter optimisation, a process to tweak internal variables such as learning rate, network architecture or regularisation in order to obtain better performance. This process invariably involves repeatedly testing the predictor's performance against a certain "test" data set and has potential to introduce another source of bias, even into unsupervised methods. Our use of DMS data to assess these methods should greatly reduce the impact of this effect for all methods except Envision and possibly DeepSequence, which could have conceivably been optimised against DMS data used in its original evaluation. Envision does not perform exceptionally regardless, and we show that DeepSequence still performs well when assessed using data it has definitely not seen (Table EV9).

Certain DMS experiments appear to show outstanding performance at identifying disease mutations. It is interesting to compare performance with respect to the experimental phenotypes used, as the utility of an experimental phenotype for identifying pathogenic mutations should be related to the mechanism by which mutations cause disease. We note that those DMS experiments based upon competitive growth assays appear to perform particularly well, ranking above all computational predictors for three of the four proteins where they are available. For BRCA1, where there are DMS data sets based upon three different experimental phenotypes, the growth rate-based assay (Findlay *et al*, 2018) performs much better than those based upon yeast two-hybrid or E3 ubiquitin ligase activity (Starita *et al*, 2015). Growth rate is likely to be a very general experimental phenotype that will reflect any loss of function occurring at a molecular level. In contrast, if some of the pathogenic BRCA1 mutations acted by some mechanism other than perturbation of its interaction with specific binding partners (BARD1) or disrupting E3 activity, this could explain the underperformance of the DMS data based upon these alternate phenotypes. Interestingly, however, the HRAS DMS data, which is also superior to all computational predictors, are based upon a two-hybrid probe of its interaction with RasGAP (Bandaru *et al*, 2017), suggesting that disruption of this interaction is reflective of the molecular mechanisms underlying disease.

PTEN is also noteworthy, as it too has different DMS data sets available based upon different experimental phenotypes. The screen for the PTEN(b) data set assesses the disruption of an artificial gene circuit in yeast, essentially probing phosphatase activity. This data set is superior to all but four VEPs, suggesting it is reasonably reflective of molecular disease mechanisms. In contrast, the phenotypic screen for PTEN(a) measures protein abundance in the cell by fluorescence of EGFP bound to the protein (Matreyek *et al*, 2018). This technique, called VAMP-seq, identifies thermodynamically unstable variants; however, this may fail to capture disease mechanisms acting through interaction disruption and loss or gain of function unrelated to destabilisation. Indeed, it was noted in this study that dominant-negative variants were not significantly different from wild type, consistent with our previous observation that dominant-negative mutations tend to be very mild at the protein structural level (McEntagart *et al*, 2016). Thus, great care must be taken when selecting an experimental phenotype. In the absence of a better phenotypic assay specifically related to a known disease mechanism, experiments based upon growth may be the most general way of probing loss of protein function, and thus the most useful for predicting disease.

Our results in analysing the predictive capability of DMS data sets largely recapitulate the results presented in the original studies. The CALM1 data set (Weile *et al*, 2017) is reported to have superior precision–recall performance than PolyPhen-2 and PROVEAN, which we also find (for the raw scores rather than the flipped scores). The TPK1 data set (Weile *et al*, 2017) allowed complete separation of the neutral and disease alleles as did Poly-Phen-2 and PROVEAN, but only after additional filtering for recessive disease alleles, which we did not perform. The BRCA1(a) data set (Starita *et al*, 2015) is used by the authors to train a model to predict homology-direct DNA repair rescue; however, predictions are primarily made outside of the region of DMS coverage which we are unable to assess. BRCA1(b) (Findlay *et al*, 2018) is reported by the authors to separate pathogenic and benign mutations in ClinVar almost perfectly, a result which we also see in our analysis. The PTEN(a) (Matreyek *et al*, 2018) data set is stated to identify upwards of 90% of PTEN pathogenic variants, although no false-positive rate is given since no PTEN variants were officially classified as benign. Again, our results are similar, given the high precision–recall AUC of the PTEN(a) data set but the considerably lower-ranked ROC AUC indicates a significant false-positive rate. Finally, the PTEN(b) authors (Mighell *et al*, 2018) employed a similar approach to us, using gnomAD variants to stand-in for benign substitutions. Their results indicate that their data have a superior positive predictive value than PROVEAN, SIFT and PolyPhen-2 which we also find.

The two most commonly used VEPs are probably PolyPhen-2 and SIFT, which are both still very widely used in variant prioritisation. Neither showed exceptional performance in this study, ranking 14[th] and 25[th] against the human DMS data (although SIFT4G, a genomic-conservation-based implementation of the SIFT algorithm (Vaser *et al*, 2016) ranked 9[th]). Therefore, we recommend other VEPs based upon our analyses. Unfortunately, Deep-Sequence is very computationally intensive and could be quite difficult for a typical end user to run. It also does not have defined disease thresholds; these would need to be assessed on a protein-by-protein basis, likely by analysis of putatively benign

(e.g. gnomAD) variants. We therefore highlight SNAP2, DEOGEN2, SNPs&GO and SuSPect, which also tended to perform well against the DMS data sets, and have simple-to-use web interfaces. We further recommend REVEL—although it lacks a web interface, it has been pre-calculated for all human chromosomes and is available online to download. We suggest that these methods would make good choices for routine variant prioritisation. Importantly, however, they all showed large variation in their performance between different proteins, suggesting that one should still not rely too much on the results of any single predictor.

While evolutionary conservation is widely accepted to be the most predictive feature used in variant effect prediction, some VEPs also integrate features derived from experimentally determined protein structures (PolyPhen-2, S3D-PROF, SNP&GOs3D, DEOGEN2 and MPC). It is interesting that the inclusion of protein structural models did not appear to be particularly useful for the VEPs. In principle, since disease mechanisms can often be explained by protein structural effects (Steward *et al*, 2003) one might expect that protein structure should be useful. It may be that the value of evolutionary information simply dwarfs any contribution from the inclusion of structure; that is, if a mutation is damaging at a structural level, this is likely to be reflected in the evolutionary conservation of that residue. Moreover, many pathogenic mutations are not highly damaging at a protein structural level, for example those associated with a dominant-negative effect in protein complexes (Bergendahl *et al*, 2019) or those that affect transcription factor binding specificity (Williamson *et al*, 2019). It is possible that future strategies that take into consideration the diverse molecular mechanisms underlying human genetic disease and the unique structural properties of individual proteins will be able to make better use of the huge amount of protein structural data now available.

The value of DMS data for directly identifying pathogenic mutations is especially exciting, based on the results we observed here. Given the proper choice of experimental phenotype, DMS experiments are likely to be better than (or at the very least competitive with) the best computational VEPs. The applicability of DMS data for direct variant prioritisation is currently limited by the small fraction of human protein residues for which DMS experiments have been performed. In the coming years, as more proteins are studied and experimental strategies are improved, we expect that the utilisation of such data for the identification of damaging variants will become routine.

## Materials and Methods

### Selecting DMS data sets for correlation analysis

Most of the DMS studies analysed provided multiple fitness maps for the protein of interest. Depending on the study, this was due to replicates in differing conditions (e.g. multiple antibiotic concentrations), different functional assays or quality filtering of the results. As our interest was to see how well VEPs could replicate the results of DMS experiments, for proteins with multiple data sets available from a single study, we selected the fitness map with the highest average Spearman correlation to all predictors to assess in Fig 2 and

Table EV7. Where a quality threshold was given, separating high- and low-quality results, we tested all results, and high-quality filtered results. We did not investigate imputed results or those generated by predictive models trained on the DMS fitness maps.

### Structure selection

The SNP&GOs3D VEP along with S3D-PROF require a protein structure to be provided in order to generate results. Where possible, we selected an X-ray crystallography structure with a resolution ≤ 2.5 Å and selected the structure with greatest coverage of the DMS results for that protein. Otherwise we selected the highest resolution structure available. A full list of the structures and chains used for these predictors is provided in Table EV14.

### Calculating rank scores

Rank score is defined as the mean, normalised correlation over all proteins, given by the following formula:

$$R = \frac{\sum_m \frac{c - c_{\min}}{c_{\max} - c_{\min}}}{m_x}$$

where $c$ is each correlation for a specific protein, $c_{\min}$ is the minimum correlation for each protein and $c_{\max}$ is the maximum correlation for each protein. This represents the correlation, normalised to a scale between 1 for the highest ranking method and 0 for the lowest. This is then summed across all proteins ($m$) for the same method, and divided by the number of proteins for which this method generated a result ($m_x$), in order to normalise for instances where a predictor failed to generate results for a certain protein. Where multiple DMS data sets are present for a single protein, we averaged the normalised correlations of each predictor between these data sets, and treated the resulting values as scores from a single protein.

Coefficient of variation is calculated from the normalised correlations, before the mean is taken. It is the standard deviation of these values across all proteins, divided by the mean. This represents the variation in predictor rank between different proteins.

It should be noted that rank scores are only comparable within the set of proteins that were used to calculate it, nor does it convey any information about predictor accuracy. The rank score metric can only be used for relative ranking within a set of proteins.

### Human mutation data sets

Data were retrieved from gnomAD v2.1 by searching for each of the human genes at https://gnomad.broadinstitute.org. Because CALM1 had only eight missense variants in gnomAD v2.1, we also included an additional three missense variants from gnomAD v3.0 (for CALM1 only). We did not filter for allele frequency. Each gene was also searched for in the ClinVar database at https://www.ncbi.nlm.nih.gov/clinvar. Data were filtered so that only missense mutations labelled as "pathogenic" or "likely pathogenic" were present.

### Plotting ROC and precision–recall curves

To plot the ROC curves, mutations present in the gnomAD data set were taken as true negatives, while mutations present in the ClinVar data set were taken as true positives. Mutations present in both sets were removed from the gnomAD set. The "roc_curve" and "auc" functions for the sklearn python package were used to calculate the true-positive rate (TPR) and false-positive rate (FPR) and the AUC. As some predictors utilise inverse metrics and thus produce an AUC under 0.5, we multiplied the predictions of all such methods by −1 to bring the value above 0.5; this is equivalent to inverting the TPR and FPR. Precision–recall curves were calculated using the "precision_recall_curve" and "auc" functions from the sklearn package. A list of methods with inverted scores was retained from the ROC calculations. The scores from these methods were deducted from one to retain comparability. As precision–recall curves are sensitive to class balance, we removed methods with less-than-complete coverage of the DMS mutations within the ClinVar and gnomAD data sets from the analysis. We also plotted individual curves for DMS assays in the same protein with differing coverage of the available ClinVar and gnomAD mutations.

### Bootstrapping

To calculate statistical significance, we utilised a bootstrapping methodology and applied it to both the VEP ranking analysis using DMS data and the ROC curve calculation. For the ranking analysis, we re-sampled mutations from each protein with replacement 1,000 times and re-calculated the rank scores. Our $P$-value for the top-ranking method was therefore the number of times it did not produce the top rank score, divided by 1,000. The ROC curve bootstrapping was carried out using the same method with 10,000 replicates except the ClinVar and gnomAD mutations were sampled individually to retain class balance and ensure that there was no chance one class could be lost from the analysis. The $P$-value for one method performing significantly better than another was the number of times it underperformed the second method, divided by 10,000.

## Data availability

The data sets containing all variant effect predictions and DMS measurements used in this study are available at Figshare.

Variants from all organisms: https://doi.org/10.6084/m9.figshare.12369359.v1

Human pathogenic and putatively benign variants: https://doi.org/10.6084/m9.figshare.12369452.v1

Expanded View for this article is available online.

### Acknowledgements

We thank Deborah Marks and Adam Riesselman for their helpful clarifications regarding DeepSequence. JM is supported by an MRC Career Development Award (MR/M02122X/1) and is a Lister Institute Research Prize Fellow. BL is supported by the MRC Precision Medicine Doctoral Training Programme.

### Author contributions

The study idea was conceived by JAM. BJL acquired all data for use in the study and carried out all data analysis. JAM supervised and aided in interpreting the results. BJL took a lead in writing the manuscript, which was finalised with considerable contributions from JAM.

## Conflict of interest

The authors declare that they have no conflicts of interest.

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
