## [Review Process File · Molecular Systems Biology]

Using deep mutational scanning to benchmark variant effect predictors and identify disease mutations

Benjamin Livesey and Joseph Marsh

DOI: [10.15252/msb.20199380](https://doi.org/10.15252/msb.20199380)

Corresponding author(s): Joseph Marsh (joseph.marsh@igmm.ed.ac.uk)

Review Timeline:

Submission Date:	27th Nov 19
Editorial Decision:	21st Jan 20
Revision Received:	8th Apr 20
Editorial Decision:	7th May 20
Revision Received:	18th May 20
Accepted:	26th May 20

Editor: Maria Polychronidou

Transaction Report:

Manuscript Number: MSB-19-9380, Using deep mutational scanning data to benchmark phenotype predictors and identify disease mutations

Thank you again for submitting your work to Molecular Systems Biology. Overall, the reviewers acknowledge that the presented findings are likely to be of interest for the genetics and clinical genetics fields. They raise however a series of concerns, which we would ask you to address in a revision.

As you will see below, the most substantial issues are raised by reviewer #3, who is concerned about circularity potentially leading to an overestimation of performance, requests statistical support for the method comparison and recommends taking into consideration how the reported scores would be interpreted in a clinical genetics setting. All other issues raised by the referees would need to be convincingly addressed.

REFEREE REPORTS

Reviewer #1:

In their manuscript "Using deep mutational scanning data to benchmark computational phenotype predictors and identify pathogenic missense mutations" Livesey and Marsh review currently available computational variant effect predictors, with a focus on amino acid substitutions, and how they predict deep mutational scanning (DMS) data, i.e. highly parallel reporter assays of protein mutations. The manuscript is well written and of general interest. I have a number of small comments (below), as well as some more general ones, which I think the authors need to address:

(1) The authors state several times, that DMS is unbiased while at the same time discussing different biases in these data sets, e.g. due to the different assays/read-outs used or technical factors like different "coverage" of certain amino acid substitutions. I would strongly advise against calling DMS data sets unbiased. They are independent from the functional read-outs that we get

from disease studies, but they are by no means unbiased either.

(2) In different sections of their manuscript, the authors discuss that commonly applied validation data sets/methods are biased. At one point, they are discussing them specifically as Type 1 and Type 2 circularity. I think Type 2 should be described much more broadly, so that it also includes ascertainment biases (e.g. study of highly conserved or long-studied genes) in the training data and shared with the validation data. A lot of studies are performed using hold-out data, however such hold-outs will always share the same biases and if the bias is part of the discriminative power of the model, the generalization abilities of the method will be overstated.

(3) I am aware of a study proposing a method called Envision which was published at the end of 2017, which looks into how DMS data sets could be used to build a predictor of variant effects. The study seems very related (assessing different tools and discussing the general ability/usefulness of DMS data). I am missing a comparison to this method and an incorporation of the conclusions from this study.

Minor:

(1) The part with "phenotype predictors" in the title seems an overstatement. None of the tested methods is aimed at "phenotype" prediction. Maybe "variant effect predictors" would be better.

(2) "Part of the mutational background", what you mean is "functionally neutral".

(3) "... most commonly used feature is evolutionary conservation and [known/existing] variation (Table 2)."

(4) While the authors excluded very incomplete data sets, I do not see any filters on data quality, like number of replicate read outs/tags/barcodes or significance of a measured effect size. I believe the authors should at least discuss that in their manuscript.

(5) The assignment of methods/predictors to categories seems very ambiguous, maybe even ad-hoc. I am uncertain about the value of it.

(6) To my knowledge, Polyphen is using structural features (derived from amino acid sequence predictions), but not the actually available structures. A statement that it would be impacted by a lack of structural data seems incorrect.

(7) When using the gnomAD data set, did you excl. singletons or more generally rare variants? What's the N (of both classes) in this AUC calculations?

(8) When discussing differences in DMS experiments and assays types, I am missing a discussion of labs (i.e. prior experience with these assays) and how recent the data sets were created.

(9) "there sure to be at least some overlap" -> "most likely"

Reviewer #2:

This is an interesting and useful study that uses deep mutational scanning datasets to benchmark the performance of different computational methods for predicting whether single amino acid mutations are detrimental and, using a limited set of human disease genes, quantifying how well DMS datasets predict human disease genes compared to the computational predictors. The results will be of high interest to many geneticists including clinical geneticists as they highlight that the most widely used methods are likely not the best.

Comments and suggestions

Variant effect predictors

There is at least one reportedly highly quality variant effect predictor that is not analysed in this study - PrimateAI: <https://doi.org/10.1038/s41588-018-0167-z> - and I would really like to know how this compares to the other methods!

DMS datasets.

There are at least two 'high profile' DMS datasets missing that should really be included: BRCA1 mutational scan that, according to the original paper, performs extremely well at classifying disease mutations <https://doi.org/10.1038/s41586-018-0461-z>
A second PTEN DMS dataset that looks like it performs well [10.1016/j.ajhg.2018.03.018](https://doi.org/10.1016/j.ajhg.2018.03.018)

DMS data quality.

The authors compare the DMS data for two proteins from different studies which is interesting. But they do not compare or attempt to quantify the data quality from each of the 29 datasets (within study variability). Some DMS datasets are simply less good datasets because of high variability between biological replicates or because of high error rates due e.g. to low sequencing counts. Ideally the authors would re-process the raw data from every study through a standard pipeline to quantify the uncertainty in the enrichment scores in each study in a comparable manner. However I understand this is a lot of work and probably beyond the scope of the current study. However, I think the authors should attempt to quantify or at least discuss the quality of the data from the different studies - they likely substantially differ. It is likely not just that certain selection assays are more or less relevant to a human disease but also that the quality of the selections and data are important.

Data availability. It would be extremely useful if the authors could make the compiled data from the 29 DMS experiments available in one place in the supplement for others to re-analyze.

ROC-AUC analysis. It would be good to also present the AUCs for precision-recall curves

Systematic biases. Are there any systematic biases in the predictions when the DMS datasets perform poorly compared to the computational methods? e.g. prediction is ok for some domains but not others?

Methods

It would be good to state the dates when the authors performed the searches for the various datasets.

Fig 2 and all similar plots. Adding 'jitter' would make it easier to compare the different classes because at the moment the points are overlaid and not visible.

Fig 4 is a table

Reviewer #3:

The manuscript "Using deep mutational scanning data to benchmark phenotype predictors and identify disease mutations" is a timely analysis of data emerging from the rapidly growing field of deep mutational scanning (DMS), in which multiplexed assays are performed on many sequence variants for a particular target gene. The study seeks to compare the ability of computational variant effect predictors to predict DMS data, and also to compare DMS with computational methods in their respective abilities to classify clinical variant pathogenicity. Each of these questions is important and holds the potential for broad interest and high impact. However, I have a few major concerns about the study, and a longer list of minor concerns.

Major concerns

1) Performance overestimation due to circularity. The authors seem well aware of this important issue, based e.g., on this text in the introduction: "...when a machine-learning method is employed, overfitting of the training set can become an issue. For this reason, machine-learning techniques are usually subject to out-of-sample validation, whereby data not present in the training set is used to verify that the predictor has indeed learned how to classify the data. Furthermore, when other datasets are used to benchmark these techniques, they should contain as few mutations used during training and validation as possible." This said, it is then surprising the study then takes so few steps to avoid this trap in their comparison of the ability of computational methods and DMS to predict clinical pathogenicity annotations. We are left wondering to what extent the 'winning' computational methods 'peek' at the test data?

The first key example of this is the DeepSequence, the 'winning' method in predicting DMS data. This method is in fact trained on many of the very DMS data sets that are used in testing, so it is perhaps unsurprising that it appears to do well at predicting the data on which it was trained. The authors should flag this issue for DeepSequence and any other computational methods for which this was the case, and only evaluate these methods using DMS data that had not been used in training.

There is much more potential for inflated performance estimates in the evaluation of computational methods for variant effect prediction, as many of the methods evaluated will have trained on the very data used here for evaluation. There are paths the study might have taken to avoid this, for example: a) only evaluate annotated variants that have certainly not been used in training, e.g., those that first appeared in the union of ClinVar, HGMD and other common training sets after the publication date of the paper (or since the last date on which the methods have been retrained post-publication); or b) Consider only computational methods that are essentially unsupervised, e.g., BLOSUM, SIFT or PROVEAN, which either use no clinical annotations or use them 'lightly' in hyperparameter estimation for a model with so few hyperparameters that memorizing the training data would be infeasible.

2) Lack of measures of statistical significance in method comparison. In predicting DMS data, whether the winning method is significantly better than other methods vs being in a statistical 'dead heat' should be determined, with appropriate corrections for multiple testing. This should also be done for methods used in predicting clinical variant interpretations. In addition, DMS should be compared with each of the higher-ranking methods, as the question of whether any computational method beats DMS will be of general interest. Where conclusions are drawn about methods that use structural features vs those that don't, or between "trained" and untrained" methods, an appropriate statistical test should be used.

3) The study should provide an analysis of the ability of computational methods to predict DMS data that is limited to those missense variants that are possible with only a single-nucleotide change. This is the ~25% of all possible missense variants that represents the vast majority of clinically-observed missense variation.

4) If a clinical geneticist were using DMS data to identify pathogenic variants, it seems obvious that they should discount any score that does not provide an estimate of uncertainty in that score. It seems equally obvious that they should also discount scores with very high error estimates, or perhaps use the score at the most conservative position within a confidence interval. By contrast, the current study seems not to give diminished weight to results from DMS maps that did not estimate error (e.g. all three of the studies praised for providing 100% coverage), nor does it seem to consider any provided error estimates in predicting pathogenicity, or in evaluating computational predictors.

Minor concerns:

Pg 1- "The vast majority of mutations identified by sequencing are 'variants of unknown significance'". This will be misleading to readers aware of the commonly accepted definition of "variant of unknown significance" as a classification in clinical genetics. The vast majority of mutations identified by sequencing have not been clinically classified. Of those that have, just over 50% (not a 'vast majority') have been classified as variants of uncertain significance".

Pg 4. "Each study also varied in the coverage of possible single amino acid substitutions across the entire protein (Figure 1), with only three projects investigating all possible mutations across their respective proteins (IF-1, HA-H3N2 and P53)." This is a bit misleading, as many more studies than this have carried out mutagenesis across the entire protein and shown that essentially all substitutions were present at some frequency in that library. However, other studies have (quite rightly) filtered out amino acid substitutions that were so poorly represented in the library that changes in allele frequency could not be accurately measured. Here the authors are singling out for praise three studies that had failed to identify and eliminate poorly-measured substitutions.

Pg 5. It might be clearer to use the more traditional terms "supervised" and "unsupervised predictors" instead of "trained" and "untrained predictors". It may also be worth noting that many so-called unsupervised or untrained methods actually do optimize hyperparameters using training data.

Pg 7. "We then calculated the area under the curve (AUC) for each plot as a measure of that predictor's accuracy in classifying the data". In classification, accuracy is generally defined as the fraction of classifications that are correct. This is not what the area under an ROC curve measures. Suggest replacing "accuracy" with "performance". Also, an ROC curve does not reflect the way that predictions are used in practice, so that precision vs recall performance would be more illuminating.

- How do results of the study compare, where DMS maps have been previously analyzed in terms of their relative ability to identify pathogenic variants? If there are differences, what might explain them?

Pg. 11. The use of random gnomAD variants as a proxy for benign variants is perhaps a necessary evil. However, it should be noted that, although gnomAD has filtered out subjects with childhood diseases likely to have a genetic cause, that gnomAD draws upon many case-control studies, so that gnomAD subjects may be MORE likely than random people to have a genetic disorder.

Response to Reviewer #1

In their manuscript "Using deep mutational scanning data to benchmark computational phenotype predictors and identify pathogenic missense mutations" Livesey and Marsh review currently available computational variant effect predictors, with a focus on amino acid substitutions, and how they predict deep mutational scanning (DMS) data, i.e. highly parallel reporter assays of protein mutations. The manuscript is well written and of general interest. I have a number of small comments (below), as well as some more general ones, which I think the authors need to address:

(1) The authors state several times, that DMS is unbiased while at the same time discussing different biases in these data sets, e.g. due to the different assays/read-outs used or technical factors like different "coverage" of certain amino acid substitutions. I would strongly advise against calling DMS data sets unbiased. They are independent from the functional read-outs that we get from disease studies, but they are by no means unbiased either.

> This is a good point, and we certainly that there some types of bias in DMS experiments. We now use the term 'independent' instead of unbiased, referring to the fact that they are independent from any training and testing data used by the phenotype predictors

(2) In different sections of their manuscript, the authors discuss that commonly applied validation data sets/methods are biased. At one point, they are discussing them specifically as Type 1 and Type 2 circularity. I think Type 2 should be described much more broadly, so that it also includes ascertainment biases (e.g. study of highly conserved or long-studied genes) in the training data and shared with the validation data. A lot studies are performed using hold-out data, however such hold-outs will always share the same biases and if the bias is part of the discriminative power of the model, the generalization abilities of the method will be overstated.

> We have now expanded our description of Type 2 data circularity to include a discussion of ascertainment bias (page 3, 3rd paragraph).

(3) I am aware of a study proposing a method called Envision which was published at the end of 2017, which looks into how DMS data sets could be used to build a predictor of variant effects. The study seems very related (assessing different tools and discussing the general ability/usefulness of DMS data). I am missing a comparison to this method and an incorporation of the conclusions from this study.

> We have now have included Envision in this study (amongst several other new predictors). There are obvious caveats with evaluating Envision against some of the same DMS datasets that were used to train it, but, perhaps surprisingly, it does not particularly stand out in terms of performance against the DMS data. We have added a paragraph about Envision to the Discussion (page 12, last paragraph).

Minor:

(1) *The part with "phenotype predictors" in the title seems an overstatement. None of the tested methods is aimed at "phenotype" prediction. Maybe "variant effect predictors" would be better.*

> **We now use the term "variant effect predictors" (VEPs) throughout the paper.**

(2) *"Part of the mutational background", what you mean is "functionally neutral".*

> **We have changed this to say "functionally neutral".**

(3) *"... most commonly used feature is evolutionary conservation and [known/existing] variation (Table 2)."*

> **We have changed this to "known variation".**

(4) *While the authors excluded very incomplete data sets, I do not see any filters on data quality, like number of replicate read outs/tags/barcodes or significance of a measured effect size. I believe the authors should at least discuss that in their manuscript.*

> **The issue of quality control is very important, but difficult to address to the large heterogeneity in experimental approaches and published datasets. We have now included a discussion of quality control. In addition, for those experiments where filtered 'high-quality' datasets are available, we have compared their results to the full available datasets. Unsurprisingly, the high-quality datasets tend to show stronger correlations with the VEPs, and superior power for predicting disease mutations (Tables EV8-9, Fig 3).**

(5) *The assignment of methods/predictors to categories seems very ambiguous, maybe even ad-hoc. I am uncertain about the value of it.*

> **This is fair, and we have now substantially improved our category assignment, using more concrete rules, as described in the *Summary of Major Changes* section.**

(6) *To my knowledge, PolyPhen is using structural features (derived from amino acid sequence predictions), but not the actually available structures. A statement that it would be impacted by a lack of structural data seems incorrect.*

> **PolyPhen uses both predicted structural features, and actual structures where available. However, we have now removed our structural category, and instead include a brief discussion of the results of predictors that do use structures.**

(7) *When using the gnomAD data set, did you excl. singletons or more generally rare variants? What's the N (of both classes) in this AUC calculations?*

> **We did not use any filter for allele frequency in our gnomAD dataset – we now explicitly state this in the Methods. The number in both classes have now been shown on the ROC and PR curves. The ClinVar and gnomAD datasets are provided in a supplemental dataset.**

(8) *When discussing differences in DMS experiments and assays types, I am missing a discussion of labs (i.e. prior experience with these assays) and how recent the data sets were created.*

> **This would be interesting, but we do not feel it is our place to discuss the expertise and experience of the different labs performing these experiments. We provide a summary of DMS datasets and experimental methods in Table 2, and have also expanded our discussion of how experimental phenotype is related to results (starting at page 13, 2nd paragraph).**

(9) "there sure to be at least some overlap" -> "most likely"

> **We have made this change.**

Response to Reviewer #2

This is an interesting and useful study that uses deep mutational scanning datasets to benchmark the performance of different computational methods for predicting whether single amino acid mutations are detrimental and, using a limited set of human disease genes, quantifying how well DMS datasets predict human disease genes compared to the computational predictors. The results will be of high interest to many geneticists including clinical geneticists as they highlight that the most widely used methods are likely not the best.

Comments and suggestions

Variant effect predictors

There is at least one reportedly highly quality variant effect predictor that is not analysed in this study - PrimateAI: <https://doi.org/10.1038/s41588-018-0167-z> - and I would really like to know how this compares to the other methods!

> **PrimateAI has now been included in our study, although its performance was not particularly good.**

DMS datasets.

There are at least two 'high profile' DMS datasets missing that should really be included:

BRCA1 mutational scan that, according to the original paper, performs extremely well at classifying disease mutations <https://doi.org/10.1038/s41586-018-0461-z>

A second PTEN DMS dataset that looks like it performs well [10.1016/j.ajhg.2018.03.018](https://doi.org/10.1016/j.ajhg.2018.03.018)

> **Both of these datasets have now been added to our analysis. Consistent with the original publication, the new BRCA1 dataset performs strikingly well at the identification of pathogenic missense mutations.**

DMS data quality.

The authors compare the DMS data for two proteins from different studies which is interesting. But they do not compare or attempt to quantify the data quality from each of the 29 datasets (within study variability). Some DMS datasets are simply less good datasets because of high variability between biological replicates or because of high error rates due e.g. to low sequencing counts. Ideally the authors would re-process the raw data from every study through a standard pipeline to quantify the uncertainty in the enrichment scores in each study in a comparable manner. However I understand this is a lot of work and probably beyond the scope of the current study. However, I think the authors should attempt to quantify or at least discuss the quality of the data from the different studies - they likely substantially differ. It is likely not just that certain selection assays are more or less relevant to a human disease but also that the quality of the selections and data are important.

> **We now discuss the issue of quality control and address it where possible, as described in our response to Reviewer 1.**

Data availability. It would be extremely useful if the authors could make the compiled data from the 29 DMS experiments available in one place in the supplement for others to re-analyze.

> We have now provided datasets that include the results of all computational predictors and DMS values for all relevant mutations.

ROC-AUC analysis. It would be good to also present the AUCs for precision-recall curves

> We have now also included an analysis using PR curves in Figure EV3. The results are largely similar to the ROC analysis.

Systematic biases. Are there any systematic biases in the predictions when the DMS datasets perform poorly compared to the computational methods? e.g. prediction is ok for some domains but not others?

> We further investigated instances where VEPs performed particularly poorly against ClinVar/gnomAD data, but well when compared with DMS. We identify and present four instances of potential domain-specific weightings (Fig EV4).

Methods

It would be good to state the dates when the authors performed the searches for the various datasets.

> We have now added access dates to Table 2 and for the ClinVar and gnomAD data.

Fig 2 and all similar plots. Adding 'jitter' would make it easier to compare the different classes because at the moment the points are overlaid and not visible.

> We have now added jitter to the plots.

Fig 4 is a table

> We have converted the previous Fig 4 to Table 3 (and now explicitly state the predictor categories, instead of showing them by colour).

Response to Reviewer #3

The manuscript "Using deep mutational scanning data to benchmark phenotype predictors and identify disease mutations" is a timely analysis of data emerging from the rapidly growing field of deep mutational scanning (DMS), in which multiplexed assays are performed on many sequence variants for a particular target gene. The study seeks to compare the ability of computational variant effect predictors to predict DMS data, and also to compare DMS with computational methods in their respective abilities to classify clinical variant pathogenicity. Each of these questions is important and holds the potential for broad interest and high impact. However, I have a few major concerns about the study, and a longer list of minor concerns.

Major concerns

1) *Performance overestimation due to circularity. The authors seem well aware of this important issue, based e.g., on this text in the introduction: "...when a machine-learning method is employed, overfitting of the training set can become an issue. For this reason, machine-learning techniques are usually subject to out-of-sample validation, whereby data not present in the training set is used to*

verify that the predictor has indeed learned how to classify the data. Furthermore, when other datasets are used to benchmark these techniques, they should contain as few mutations used during training and validation as possible." This said, it is then surprising the study then takes so few steps to avoid this trap in their comparison of the ability of computational methods and DMS to predict clinical pathogenicity annotations. We are left wondering to what extent the 'winning' computational methods 'peek' at the test data?

The first key example of this is the DeepSequence, the 'winning' method in predicting DMS data. This method is in fact trained on many of the very DMS data sets that are used in testing, so it is perhaps unsurprising that it appears to do well at predicting the data on which it was trained. The authors should flag this issue for DeepSequence and any other computational methods for which this was the case, and only evaluate these methods using DMS data that had not been used in training.

> Importantly, DeepSequence was not directly trained using DMS data. The only predictor we know of trained on DMS data is Envision, which is now included in our study, as suggested by Reviewer 1 (but we specifically discuss the potential for overfitting when assessing it using DMS data). It is possible that DeepSequence underwent some hyperparameter optimisation against the DMS data used in this study, as some of the experiments from our analyses were used in the evaluation of DeepSequence in its original publication. We now discuss the issue of hyperparameter optimisation, but conclude that it is unlikely to account for the large outperformance of DeepSequence compared to other predictors (page 12, 3rd paragraph).

There is much more potential for inflated performance estimates in the evaluation of computational methods for variant effect prediction, as many of the methods evaluated will have trained on the very data used here for evaluation. There are paths the study might have taken to avoid this, for example: a) only evaluate annotated variants that have certainly not been used in training, e.g., those that first appeared in the union of ClinVar, HGMD and other common training sets after the publication date of the paper (or since the last date on which the methods have been retrained post-publication); or b) Consider only computational methods that are essentially unsupervised, e.g., BLOSUM, SIFT or PROVEAN, which either use no clinical annotations or use them 'lightly' in hyperparameter estimation for a model with so few hyperparameters that memorizing the training data would be infeasible.

> It is crucial for us to emphasise that the primary purpose of the disease mutation analysis in Figure 3 is to compare DMS data to variant effect predictors, and not to compare the computational predictors to each other. If our goal was to compare computational predictors in distinguishing between ClinVar and gnomAD variants, we could have run it on far more proteins, but instead, we are only looking at those where DMS datasets are available. We do make some comparison between the performance of difference VEPs in discriminating between pathogenic and benign variants, but emphasise the potential issues with overfitting, and note that it is remarkable that the unsupervised DeepSequence outperforms all of the other methods. We have now attempted to make the purpose of this analysis clearer in the text (page 10, 4th paragraph).

2) Lack of measures of statistical significance in method comparison. In predicting DMS data, whether the winning method is significantly better than other methods vs being in a statistical 'dead heat' should be determined, with appropriate corrections for multiple testing. This should also be done for methods used in predicting clinical variant interpretations. In addition, DMS should be compared with each of the higher-ranking methods, as the question of whether any computational method beats DMS will be of general interest. Where conclusions are drawn about methods that use

structural features vs those that don't, or between "trained" and untrained" methods, an appropriate statistical test should be used.

> We have now included a bootstrapping approach to assess the statistical significance of top-ranking methods in both analyses. We find that DeepSequence is significantly better than all other computational predictors in terms of its correlation with experimental DMS data (page 7, 3rd paragraph). DMS data performs significantly better than all computational predictors except DeepSequence at discriminating between pathogenic and benign variants (page 10, 2nd paragraph).

3) The study should provide an analysis of the ability of computational methods to predict DMS data that is limited to those missense variants that are possible with only a single-nucleotide change. This is the ~25% of all possible missense variants that represents the vast majority of clinically-observed missense variation.

> We have now performed this analysis including only substitution possible by single nucleotide changes in Table EV6. There were some minor changes in the overall rankings, but no differences in the top-ranked methods.

4) If a clinical geneticist were using DMS data to identify pathogenic variants, it seems obvious that they should discount any score that does not provide an estimate of uncertainty in that score. It seems equally obvious that they should also discount scores with very high error estimates, or perhaps use the score at the most conservative position within a confidence interval. By contrast, the current study seems not to give diminished weight to results from DMS maps that did not estimate error (e.g. all three of the studies praised for providing 100% coverage), nor does it seem to consider any provided error estimates in predicting pathogenicity, or in evaluating computational predictors.

> As mentioned in response to both previous reviewers, we have now addressed the issue of quality control, and done some comparison of 'high-quality filtered' vs full datasets where possible. While it would be interesting to directly incorporate error estimates into our analyses, this is very difficult to the extreme heterogeneity in the way different experiments were performed and resulting datasets made available. We certainly agree that it is a critical issue to address if DMS data is to be routinely used to aid clinical diagnoses.

Minor concerns:

Pg 1- "The vast majority of mutations identified by sequencing are 'variants of unknown significance'". This will be misleading to readers aware of the commonly accepted definition of "variant of unknown significance" as a classification in clinical genetics. The vast majority of mutations identified by sequencing have not been clinically classified. Of those that have, just over 50% (not a 'vast majority') have been classified as variants of uncertain significance".

> This sentence has been changed to talk about 'unknown phenotypic consequence'.

Pg 4. "Each study also varied in the coverage of possible single amino acid substitutions across the entire protein (Figure 1), with only three projects investigating all possible mutations across their respective proteins (IF-1, HA-H3N2 and P53)." This is a bit misleading, as many more studies than this have carried out mutagenesis across the entire protein and shown that essentially all substitutions were present at some frequency in that library. However, other studies have (quite rightly) filtered out amino acid substitutions that were so poorly represented in the library that changes in allele frequency could not be accurately measured. Here the authors are singling out for praise three studies that had failed to identify and eliminate poorly-measured substitutions.

> We removed our specific mention of these studies to avoid any impression of ‘praising’. The point was to simply describe Figure 1 in the text rather than draw attention to specific studies. It is beyond the scope of our study to perform filtering of DMS results in this way. We have now discussed data quality and filtering carried out by the DMS authors in more detail.

Pg 5. It might be clearer to use the more traditional terms "supervised" and "unsupervised predictors" instead of "trained" and "untrained predictors". It may also be worth noting that many so-called unsupervised or untrained methods actually do optimize hyperparameters using training data.

> This is a great suggestion, and we have now updated all our categories and assignment criteria, as described in the *Summary of Major Changes* section.

Pg 7. "We then calculated the area under the curve (AUC) for each plot as a measure of that predictor's accuracy in classifying the data". In classification, accuracy is generally defined as the fraction of classifications that are correct. This is not what the area under an ROC curve measures. Suggest replacing "accuracy" with "performance". Also, an ROC curve does not reflect the way that predictions are used in practice, so that precision vs recall performance would be more illuminating.

> We have now included precision-recall curves in Fig EV3, as also requested by Reviewer 2. We have also corrected the sentence to use the term ‘performance’.

- How do results of the study compare, where DMS maps have been previously analyzed in terms of their relative ability to identify pathogenic variants? If there are differences, what might explain them?

> In the Discussion section, we now include a comparison of our gnomAD/ClinVar analysis to the author’s own analysis in their paper where available (page 14, 1st paragraph). We find that our results correspond well with the original authors’ analysis of their predictive ability.

Pg. 11. The use of random gnomAD variants as a proxy for benign variants is perhaps a necessary evil. However, it should be noted that, although gnomAD has filtered out subjects with childhood diseases likely to have a genetic cause, that gnomAD draws upon many case-control studies, so that gnomAD subjects may be MORE likely than random people to have a genetic disorder

> This is a good point, and we fully agree that using gnomAD variants as putatively benign is a ‘necessary evil’. Unfortunately, however, we don’t believe that we have a better way to do it at this point in time (e.g. using evolutionary variation as benign, as has been done in the past, is problematic because so many predictions directly utilise multiple sequence alignments).

Manuscript Number: MSB-19-9380R

Title: Using deep mutational scanning to benchmark variant effect predictors and identify disease mutations

Thank you for sending us your revised manuscript. We have now heard back from the two reviewers who were asked to evaluate your study. As you will see below, both reviewers think that the study has improved as a result of the performed revisions. However, reviewer #3 points out that two of their previously raised concerns have not been satisfactorily addressed. We would therefore ask you to address these remaining concerns in an exceptional second round of revision.

On a more editorial level, we would ask you to address a few remaining editorial issues listed below.

REFeree REPORTS

Reviewer #2:

The authors have addressed my concerns. This is an interesting and useful resource for the human genetics, genetic prediction and deep mutagenesis fields.

Reviewer #3:

Review of first revision of "Using deep mutational scanning to benchmark variant effect predictors and identify disease mutations"

The revised manuscript is much improved, and I continue to think that the subject matter of the manuscript is both important and timely, and that a substantial subset of the analysis is quite interesting and important. However, the revision failed to fully address two major and quite addressable issues that were raised previously. These issues undermine stated conclusions in at least two major areas: 1) claims that the performance of DeepSequence in predicting DMS data exceeds that of all other methods; 2) comparisons amongst other computational methods in terms of predicting pathogenicity.

Major concerns

1- My first major concern from the first review, stated again here, applies equally well to the revision (discussed further below the authors' response).

Initial Reviewer concern: "Performance overestimation due to circularity. The authors seem well aware of this important issue, based e.g., on this text in the introduction: "...when a machine-learning method is employed, overfitting of the training set can become an issue. For this reason, machine-learning techniques are usually subject to out-of-sample validation, whereby data not present in the training set is used to verify that the predictor has indeed learned how to classify the data. Furthermore, when other datasets are used to benchmark these techniques, they should contain as few mutations used during training and validation as possible." This said, it is then surprising the study then takes so few steps to avoid this trap in their comparison of the ability of computational methods and DMS to predict clinical pathogenicity annotations. We are left wondering to what extent the 'winning' computational methods 'peek' at the test data?

The first key example of this is the DeepSequence, the 'winning' method in predicting DMS data. This method is in fact trained on many of the very DMS data sets that are used in testing, so it is perhaps unsurprising that it appears to do well at predicting the data on which it was trained. The authors should flag this issue for DeepSequence and any other computational methods for which this was the case, and only evaluate these methods using DMS data that had not been used in training."

Authors' response:

"Importantly, DeepSequence was not directly trained using DMS data. The only predictor we know of trained on DMS data is Envision, which is now included in our study, as suggested by Reviewer 1 (but we specifically discuss the potential for overfitting when assessing it using DMS data). It is possible that DeepSequence underwent some hyperparameter optimisation against the DMS data used in this study, as some of the experiments from our analyses were used in the evaluation of DeepSequence in its original publication. We now discuss the issue of hyperparameter optimisation, but conclude that it is unlikely to account for the large outperformance of DeepSequence compared to other predictors (page 12, 3rd paragraph)."

Further support for the seriousness of this concern:

I encourage the authors to re-examine the DeepSequence Nature Methods paper, specifically a section called "Bias correction" near the end of the Methods section. Here it is made clear that the DeepSequence prediction score is the sum of two scores, one based on the deep learning method emphasized in the main text and another based on (explicitly supervised) regression, that was

directly trained on DMS data. Thus, DeepSequence is in fact based on supervised learning trained on precisely the same data DMS that the present study uses to judge performance. The current study's performance estimate of DeepSequence in predicting DMS may well be artificially inflated. Indeed, the fact that the UBI4 DMS dataset on which DeepSequence performs least well was not listed amongst DMS datasets used in this paper further supports inflated performance for the DMS datasets they used. This finding should be removed altogether or restricted to newer DMS datasets that were not used in training DeepSequence (those used are listed in their Supplementary Table 2).

1- My second major concern, raised in the first review and stated again here, still applies to the revision.

From my initial review:

"There is much more potential for inflated performance estimates in the evaluation of computational methods for variant effect prediction, as many of the methods evaluated will have trained on the very data used here for evaluation. There are paths the study might have taken to avoid this, for example: a) only evaluate annotated variants that have certainly not been used in training, e.g., those that first appeared in the union of ClinVar, HGMD and other common training sets after the publication date of the paper (or since the last date on which the methods have been retrained post-publication); or b) Consider only computational methods that are essentially unsupervised, e.g., BLOSUM, SIFT or PROVEAN, which either use no clinical annotations or use them 'lightly' in hyperparameter estimation for a model with so few hyperparameters that memorizing the training data would be infeasible."

Authors' response:

"It is crucial for us to emphasise that the primary purpose of the disease mutation analysis in Figure 3 is to compare DMS data to variant effect predictors, and not to compare the computational predictors to each other. If our goal was to compare computational predictors in distinguishing between ClinVar and gnomAD variants, we could have run it on far more proteins, but instead, we are only looking at those where DMS datasets are available. We do make some comparison between the performance of difference VEPs in discriminating between pathogenic and benign variants, but emphasise the potential issues with overfitting, and note that it is remarkable that the unsupervised DeepSequence outperforms all of the other methods. We have now attempted to make the purpose of this analysis clearer in the text (page 10, 4th paragraph)."

Further support for the seriousness of this concern:

Whether or not it is "the primary purpose" of the paper, the paper currently compares computational predictors on the basis of pathogenic variant test data that may have been used in training (of special concern for supervised methods and metapredictors). The fact that performance of these predictors is likely inflated by circularity is not excused by the author's defense that they only made the comparison based on a few proteins, as this only reduces the value of the comparison further. Of course, evaluating computational methods against annotated pathogenic variation is fine where the method (e.g., DeepSequence) has not make use of pathogenic variation as training data. For the supervised computational methods, a more rigorous comparison has many challenges, and may require using only pathogenic and presumed-benign variants that were not used directly or indirectly in training. The challenges of such a comparison serve to accentuate the elegance and great value of the author's other approach of using DMS data as an independent test set, as this test set has not been used by any of the computational methods beyond DeepSequence. Comparison of predictors against DMS data using pathogenic variation is fine, if appropriate caveats are given for computational methods, given that DMS won despite being at a disadvantage. Comparison amongst computational predictors is more fraught.

Minor:

-Pg 12 "DeepSequence ... was the top predictor of human disease mutations." This should read "top computational predictor". Other statements in that paragraph will require more extensive changes based on concerns above.

-Pg 12 " For example, some of the DMS datasets used in this study were used to evaluate DeepSequence in the original study, and it is conceivable that some level of hyperparameter optimisation influenced its success here. Analysis of this bias is beyond the scope of this study, but we regard it as relatively minor compared to training biases of supervised methods." The latter comment is dangerous, as over-fitting due to hyperparameter optimization can be quite severe where the models are highly complex and there are many features. This is the case for DeepSequence.

Conclusion: I continue to think that this a timely and important paper that will be of broad interest, once the issues above have been addressed.

Response to Reviewer #3

The revised manuscript is much improved, and I continue to think that the subject matter of the manuscript is both important and timely, and that a substantial subset of the analysis is quite interesting and important. However, the revision failed to fully address two major and quite addressable issues that were raised previously. These issues undermine stated conclusions in at least two major areas: 1) claims that the performance of DeepSequence in predicting DMS data exceeds that of all other methods; 2) comparisons amongst other computational methods in terms of predicting pathogenicity.

Major concerns

1- My first major concern from the first review, stated again here, applies equally well to the revision (discussed further below the authors' response).

Further support for the seriousness of this concern:

I encourage the authors to re-examine the DeepSequence Nature Methods paper, specifically a section called "Bias correction" near the end of the Methods section. Here it is made clear that the DeepSequence prediction score is the sum of two scores, one based on the deep learning method emphasized in the main text and another based on (explicitly supervised) regression, that was directly trained on DMS data. Thus, DeepSequence is in fact based on supervised learning trained on precisely the same data DMS that the present study uses to judge performance. The current study's performance estimate of DeepSequence in predicting DMS may well be artificially inflated. Indeed, the fact that the UBI4 DMS dataset on which DeepSequence performs least well was not listed amongst DMS datasets used in this paper further supports inflated performance for the DMS datasets they used. This finding should be removed altogether or restricted to newer DMS datasets that were not used in training DeepSequence (those used are listed in their Supplementary Table 2).

>We were surprised by the reviewer's claim that DeepSequence has been directly trained on some of the same DMS datasets we used in our analysis. This would indeed be a big problem for our analysis. From our reading of the DeepSequence paper, we believed that the "Biased correction" section referred to another analysis, and was not related to the actual output of DeepSequence predictions. To clarify this, we contacted the DeepSequence authors, Deborah Marks and Adam Riesselman, who quickly responded:

"You are correct in your understanding of DeepSequence: the predictions made by the model and presented in the main results of the paper are entirely unsupervised. The DeepSequence algorithm is trained only on a multiple sequence alignment, and model parameters are not altered in any way by DMS data.

In the "Bias Correction" section of the paper, we explore the observation that there are systematic amino acid biases in the predictions from the three different evolutionary models (independent, EVmutation, and DeepSequence) built from the same multiple sequence alignments, as shown in Supp Figure 3. In Supp Figure 4, we provide a simple leave-one-protein-out strategy to correct this bias with linear regression, but we leave the development of these sorts of techniques to future work."

We hope that the truly unsupervised nature of DeepSequence is now clear. However, to address the possibility that its parameters have been indirectly optimised against the DMS data, we also

included a new analysis that considered only DMS datasets from proteins that weren't included in the original DeepSequence paper (Table EV9). Notably, DeepSequence ranks 1st, further demonstrating that our results are not due to overfitting or data circularity.

1- My second major concern, raised in the first review and stated again here, still applies to the revision.

Further support for the seriousness of this concern:

Whether or not it is "the primary purpose" of the paper, the paper currently compares computational predictors on the basis of pathogenic variant test data that may have been used in training (of special concern for supervised methods and metapredictors). The fact that performance of these predictors is likely inflated by circularity is not excused by the author's defense that they only made the comparison based on a few proteins, as this only reduces the value of the comparison further. Of course, evaluating computational methods against annotated pathogenic variation is fine where the method (e.g., DeepSequence) has not make use of pathogenic variation as training data. For the supervised computational methods, a more rigorous comparison has many challenges, and may require using only pathogenic and presumed-benign variants that were not used directly or indirectly in training. The challenges of such a comparison serve to accentuate the elegance and great value of the author's other approach of using DMS data as an independent test set, as this test set has not been used by any of the computational methods beyond DeepSequence. Comparison of predictors against DMS data using pathogenic variation is fine, if appropriate caveats are given for computational methods, given that DMS won despite being at a disadvantage. Comparison amongst computational predictors is more fraught.

>This is a good point, and we agree it is difficult to make a fair comparison between supervised and metapredictors, given that they are very likely to have been trained on some of the mutations used in our evaluation. Therefore, we have removed any discussion of the relative performance of these predictors. It should be clear now that the primary purpose of this analysis is to highlight the remarkable performance of experimental DMS data for the identification of pathogenic mutations. However, we still note the strong performance of DeepSequence, given that it is an unsupervised method that has definitely not been trained against these mutations.

Minor:

-Pg 12 "DeepSequence ... was the top predictor of human disease mutations." This should read "top computational predictor". Other statements in that paragraph will require more extensive changes based on concerns above.

>This has been changed

-Pg 12 " For example, some of the DMS datasets used in this study were used to evaluate DeepSequence in the original study, and it is conceivable that some level of hyperparameter optimisation influenced its success here. Analysis of this bias is beyond the scope of this study, but we regard it as relatively minor compared to training biases of supervised methods." The latter comment is dangerous, as over-fitting due to hyperparameter optimization can be quite severe where the models are highly complex and there are many features. This is the case for DeepSequence.

>We have changed this so that we no longer appear to downplay the risks of hyperparameter optimisation, and we also mention our new analysis that includes only DMS datasets from proteins that weren't included in the original DeepSequence study (pg 12, last paragraph).

Conclusion: I continue to think that this a timely and important paper that will be of broad interest, once the issues above have been addressed.

Manuscript number: MSB-19-9380RR

Title: Using deep mutational scanning to benchmark variant effect predictors and identify disease mutations

Thank you again for sending us your revised manuscript and for performing the requested changes. We are now satisfied with the modifications made and I am pleased to inform you that your paper has been accepted for publication.

Corresponding Author Name: Joseph A Marsh

Manuscript Number: MSB-19-9380